# Seasonal, Organ-, and Location-Dependent Variations in the Alkaloid Content of *Pachysandra terminalis* Investigated by Multivariate Data Analysis of LC-MS Profiles

**DOI:** 10.3390/plants14193060

**Published:** 2025-10-03

**Authors:** Lizanne Schäfer, Jandirk Sendker, Thomas J. Schmidt

**Affiliations:** University of Münster, Institute of Pharmaceutical Biology and Phytochemistry (IPBP), PharmaCampus Corrensstraße 48, D-48149 Münster, Germany; l_scha57@uni-muenster.de (L.S.); sendkej@uni-muenster.de (J.S.)

**Keywords:** *Pachysandra terminalis*, Buxaceae, aminosteroid alkaloids, metabolite profile, seasonal profile, multivariate data analysis, principal component analysis, volcano plot

## Abstract

*Pachysandra terminalis* (*P. terminalis*), a plant belonging to the Buxaceae family, is known as a great source of aminosteroid alkaloids. In a previous communication, we reported on the isolation of a variety of aminosteroids from *P. terminalis,* which presented interesting activity against the protozoan pathogens, *Trypanosoma brucei rhodesiense* and *Plasmodium falciparum*. In the present study, variations in the alkaloid profile of *P. terminalis* related to seasonal changes as well as differences between plant organs (leaves and twigs) and between plant populations were investigated to prioritize candidates for targeted isolation in further studies. For this purpose, sample material of *P. terminalis* was collected from the two nearby populations in monthly intervals over one year. The ethanolic (75%) extracts were analyzed using UHPLC/+ESI-QqTOF-MS/MS, and the resulting data converted to variables encoding the intensity of MS signals in particular *m/z* and retention time (*t*_R_) intervals over the chromatographic runs. The very large and complex data matrix of these <*t*_R_:*m/z*> variables was evaluated using multivariate data analysis, especially principal component analysis (PCA) and volcano plot analysis of *t*-test data. The results of these analyses, for the first time, allowed a holistic analysis of variation in the alkaloid profiles in *P. terminalis* organs over the vegetation period. The evaluation of the PCA scores and loadings plots of principal components 1 through 3, as well as of volcano plots, highlighted 25 different compounds, mostly identified as aminosteroid alkaloids, that were most relevant for the differences between leaves and twigs and between the two populations and mainly determined the changes in their chemical profiles over the vegetation period.

## 1. Introduction

*Pachysandra terminalis* Sieb. et Zucc. (*P. terminalis*; Carpet Box, Japanese Spurge) belongs to the family of Buxaceae and is known as a rich source of bioactive alkaloids, mainly of pregnane-type steroid alkaloids [1,2,3,4,5,6,7]. Our ongoing study on that plant revealed that aminosteroids isolated from leaves and twigs of *P. terminalis* possess promising activity against the protozoan pathogenic parasites *Plasmodium falciparum* (responsible for Malaria) and *Trypanosoma brucei rhodesiense* (East African Human Trypanosomiasis) [8,9]. As already shown in some other projects on plants containing alkaloids such as *Mitragyna speciosa* (Rubiaceae) [10], *Leucojum aestivum* (Amaryllidaceae) [11], *Alstonia scholaris* (Apocynaceae) [12], as well as *Buxus sempervirens* (Buxaceae, i.e., a close relative of *P. terminalis*) [13], the alkaloid content and composition are often variable and depend on a variety of factors such as season, location, and plant organ. Our recent study on two different varieties of *Buxus sempervirens* (Buxaceae) [13] using multivariate data analysis (MVDA) of ultra high performance liquid chromatography/ electrospray ionization quadrupole time-of-flight tandem mass spectrometry (UHPLC/+ESI-QqTOF-MS/MS) data revealed 18 compounds that were mainly responsible for the chemical differences between the plant organs (leaves and twigs), the varieties *arborescens* and *suffruticosa* as well as for the seasonal changes [13].

Variations in the alkaloid profile of *P. terminalis*, to the best of our knowledge, were not investigated in this manner before. The current study was therefore conducted in order to carry out, for the first time, a systematic and holistic investigation of the alkaloid profile of the title plant, taking into account the variation between the main aerial organs, leaves, twigs, and flowers during the blossom period, along with their dynamics over one full year of vegetation. In order to take into account possible differences between different populations growing under near-identical climatic conditions, samples were obtained from two separate populations growing in different environments within the same city. Samples of leaves and twigs were obtained from both populations in monthly intervals over one year, dried, and stored until extraction. Finally, they were analyzed by UHPLC/+ESI-QqTOF-MS/MS, and the resulting data were evaluated by multivariate statistical tools such as bucket statistic plots, principal component analysis (PCA), and volcano plot analysis of *t*-test results in order to obtain insights into the main chemical determinants of variation in the alkaloid profiles.

## 2. Results

### 2.1. Sampling, LC/MS-Analysis, and Pretreatment for Data Analysis and Statistical Methods Used

Samples were taken in monthly intervals (February 2023–January 2024) from the different plant organs (leaves; twigs; flowers if present) of two separate plant populations of *P. terminalis* originating from nearby locations, termed A and B (exact location see Materials and Methods and Appendix A), about 700 m apart from each other within the same city (Münster, Germany). The plant material was air-dried and extracted with 75% ethanol. A total of 50 sample extracts were analyzed by UHPLC/+ESI-QqTOF-MS/MS (henceforth abbreviated “LC/MS”). The resulting LC/MS data were converted to a matrix of variables encoding the intensity of particular MS signals in particular *m/z* intervals within certain retention time (*t*_R_) intervals over the chromatographic runs (<*t*_R_:*m/z* variables>; also termed “buckets”). The obtained data matrix consisted of 1286 variables for 130 samples. Multivariate data analysis, such as bucket statistic plots, principal component analysis (PCA), and volcano plot analysis of *t*-test results, was applied to extract relevant information of various kinds from the complex matrix (“bucket table”) of chemical data.

### 2.2. Optimum Harvesting Periods for Previously Isolated Pachysandra Alkaloids

In our previous study, 20 alkaloids (**1–20**; for their structures see Appendix A) were isolated from the aerial parts of *P. terminalis* [9]. Of these compounds, 18 were known as natural constituents of *P. terminalis,* while two (compounds **6** and **9**) were found to be artifacts formed during the workup of the plant extract [9] and are therefore not further considered in the present work. Based on their accurate mass and retention time, <*t*_R_:*m/z*> variables representing native compounds could be localized in the bucket table, and their relative amounts in the different samples could be approximately assessed. To compare the relative contents of these individual compounds throughout the year, bucket statistic plots were inspected, which report the intensity values for a particular variable (bucket) in all samples. As an example, the bucket statistic view of pachysamine A (4.08 min:*m/z* 181.1847) is shown in Figure 1. The content of pachysamine A (compound **8**), on average, is a bit higher in the plants of location A (dots) compared to plants of location B (triangles), and the content in the twigs (green) is somewhat higher than in the leaves (red). Conspicuously, in the leaves of both populations, the highest amount of that compound is reached in May (5). Overall, it is the highest in that month in the leaves of plants from location A. A similar but less pronounced peak is observed for the concentration of **8** in the twigs within the interval (March)–April–May. Overall, it becomes clear from this plot that the optimum time to harvest in order to isolate this pachysamine A (**8**) under the given geographic/climatic circumstances would be springtime, i.e., April–May.

In the same manner, the optimum harvesting periods were determined for each of the *Pachysandra* alkaloids previously isolated by us [9]. The results are listed in Table 1.

All of the *Pachysandra* alkaloids previously isolated by us showed higher amounts in plants from location A, or, in few cases, showed approximately the same amounts in both locations. The optimum harvesting month yielding the highest relative amounts for many compounds was October (**1**, **3**, **4**, **5**, **10**, **11**, **14–18**). Only few compounds showed their optimum yields in the winter months (**2** and **13**) in twigs of *P. terminalis*. Compounds that showed their maximum yields in the spring and summer months (**7**, **8**, **19,** and **20**) were contained in larger quantities in the leaves compared to the twigs of *P. terminalis*. An overview of the optimum harvesting periods for these *Pachysandra* alkaloids and the corresponding weather data during the sampling year is shown in Figure 2. It cannot be assessed whether there is a causal relationship underlying correlations between the accumulation of certain substances in certain months and the weather conditions, but it may be allowed to mention a few such obvious correlations based on the available data. Thus, the accumulation of compound **7** is correlated with a high amount of sunshine in June 2023 (303 h) as well as with a low amount of rain and a high average temperature of 20 °C. In contrast, the accumulation of many compounds (see above) in October 2023 appears related to a high amount of precipitation (136 mm) and the associated higher irrigation intensity, accompanied by shorter sunlight and lower temperature. The observation that the amounts of most alkaloids in October are highest may have to do with the fact that they had time to accumulate over the earlier months of warmer weather.

### 2.3. Mass Spectrometric Characterization of Pachysandra Aminosteroids: Characteristic Fragments

To further characterize interesting compounds from the principal component analysis (PCA) and the volcano plots of the seasonal profile of *P. terminalis* beyond the molecular formula, characteristic fragments of their MS^2^ spectra were used for identification (see Appendix A). Musharraf et al. (2012) described characteristic fragments of the steroidal skeleton for *Sarcococca* alkaloids (also pregnane-type alkaloids) based on the number of substituents as well as on the saturation of the basic skeleton [15]. Many of these characteristic fragments were also found with the *Pachysandra* alkaloids analyzed by Flittner et al. (2021), who further extended the list of known fragments [8]. They were also found in the fragment spectra of the *Pachysandra* alkaloids previously isolated by the authors of the present study [9]. The characteristic fragments of the steroidal skeleton and some substituents are shown in Figure 3.

### 2.4. Principal Component Analysis (PCA)

PCA is a useful method to reduce the high-dimensional information content of a highly complex set of variables, such as the present bucket table, to a new set of uncorrelated variables (latent variables or principal components = PC). Due to their orthogonality and non-redundance, the latent PC variables can render the bucket table’s relevant information more effectively than the original manifest variables, so that fewer variables are necessary to render the same amount of information. Samples can be visualized in a scores plot, showing their positions in the new PC space. Similarly, the loadings plot shows which original variables contribute most to each PC. Variables with high loadings strongly influence sample placement, helping to identify key compounds by examining the mass spectra of the most influential <*t*_R_:*m/z*> variables [16].

Calculation of a principal component analysis (PCA) for the present complex bucket table allowed visualizing the characteristic differences between the sample groups in the score plots, and the variables representing compounds mainly responsible for these differences could be localized in the corresponding loadings plot. Retrieval and analysis of the underlying mass spectra from the original LC-MS raw data then allowed characterizing and—in many cases—identifying the compounds by the characteristics outlined in Section 2.3. The PCA results described are based on Pareto-scaled data. This scaling method was previously found suitable for the evaluation of the seasonal profile of *Buxus sempervirens* described by Szabó and Schmidt (2022) [13]. This type of scaling has the advantage of assigning a somewhat higher weight to variables with low abundance (e.g., minor constituents), which may show meaningful differences between sample groups that can be obscured by stronger effects of highly abundant variables (main constituents) in the case of unscaled use of variables. On the other hand, in cases such as the present one, it is more suitable than unit variance scaling, which assigns equal weight to all variables. Pareto scaling partly retains differences in variable intensity (i.e., high- vs. low-abundance constituents), which may represent important information but is completely lost when the data is scaled to unit variance. For a comparison of the PCAs calculated with different scaling methods, see Appendix A. The scores and loadings plot of PC2 plotted against PC1 of the Pareto-scaled data are shown in Figure 4.

The scores plot of PC2 vs. PC1 showed a clear distinction of the different sample groups. The quality control (QC_mix_), a mixture of all samples in equal parts, should appear in the middle of the scores plot, which is indeed the case. The loadings plot shows single buckets that are projected in the coordinate system of the principal components. Buckets that are located on the edge of the point cloud have a bigger influence on one of the PCs and are, therefore, characteristic of one of the sample groups.

#### 2.4.1. Principal Components 1 and 2: Differences in the Alkaloid Profile of the Plant Organs and Populations from Distinct Locations

The scores plot of principal component 2 (PC2) against PC1 is dominated by differences between the plant organs and the two different populations. PC1 represents the differences between the plant organs and accounts for the largest fraction (45%) of the explained variance of the data set. PC2 explains a further 16% of the variance, caused by differences between the two distinct populations. The scores plot of PC2 vs. PC1 thus already describes 61% of the total variance. The corresponding loadings plot of PC2 vs. PC1 (see Figure 5) allowed the identification of major characteristic compounds of the plant organs and different populations/locations. In correlation with the scores plot, buckets that are characteristic of the leaves appear on the left side of the loadings plot, and buckets that are characteristic of the twigs appear on the right side. Consistently, buckets occurring near the bottom are characteristic of the plants from location B, and those at the top are characteristic of plants from location A. For a closer characterization of the buckets, the molecular formula was obtained from the accurate mass (usually representing the [M + H]^+^ ion, in case of compounds with two basic amino groups, also the doubly charged [M + 2H]^2+^ ion) and compared with the previously isolated *Pachysandra* alkaloids. Spiropachysine (**15**) (12.43 min:*m/z* 463.3723) was, thus, identified as a characteristic compound of the leaves of *P. terminalis* and epipachysamine A (**14**) (8.00 min:*m/z* 403.3704) as well as 9-(*N*,*N*-Dimethylamino)-5-megastigmen-1-one (**1**) (2.50 min:*m/z* 238.2163) as compounds characteristic of the plants from location A. The molecular formulas of unidentified buckets were compared with compounds previously isolated by other authors from *Pachysandra* species [17]. For the identification of the buckets, the fragment spectra were analyzed (see Figure 6 and Appendix A) and compared with the characteristic fragments described in 2.3 (see Figure 3). Epipachysandrine A [18] (12.21 min:*m/z* 467.3666) could be identified as another compound characteristic of the leaves, and pachysandrine B [2] (11.71 min:*m/z* 501.4081) as a characteristic compound of the twigs of *P. terminalis*. Pachystermine C [19] (12.11 min:*m/z* 475.3928) turned out to be characteristic of the plants of location B.

#### 2.4.2. Principal Component 3: Seasonal Differences in the Alkaloid Profile of *Pachysandra terminalis*

The seasonal differences between the various samples could be visualized by plotting PC3 against PC1 (see Figure 7). PC3 accounts for 7% of the variance of the total data set. The location of the data points along PC3 shows an obvious dependence on the months of sampling. The samples were grouped into warm and cold months based on the average temperature of each month, as determined by the “Deutscher Wetterdienst” [14]. Months with an average temperature < 14 °C were characterized as “cold months” (October to April), and months with an average temperature ≥ 14 °C (May to September) were characterized as “warm months”. The corresponding loadings plot revealed characteristic compounds of the cold and warm months, as shown in Figure 8. Buckets that are located at the top of the loading plot are characteristic of the warm months (except for August and September), and buckets at the bottom are characteristic of the cold months (especially October to January). The previously isolated pachysamine A (**8**) (4.08 min:*m/z* 181.1847), pachysamine H [20] (12.57 min:*m/z* 465.3863), and pachysamine B [21] (12.42 min:*m/z* 443.4016) were characteristic of the warm months. Two unknown aminosteroids with the molecular formula C_24_H_44_N_2_O and C_29_H_48_N_2_O_3_ appear to be characteristic of the cold months. These compounds would be interesting targets for further isolation studies, as they are probably unknown natural compounds.

### 2.5. Pairwise Comparison of Sample Groups Using Volcano Plots

The first three principal components, PC1–PC3, thus revealed main compounds characteristic of the plant organs, the two populations, and the seasonal variations, respectively. To identify differences in compounds that are only present in the samples in low concentrations and hence are difficult to localize due to their small contributions to the model, pairwise comparison of sample groups was performed using volcano plots. Volcano plots are based on a *t*-test comparing these sample groups with respect to the individual variables’ values. The resulting *p*-values (i.e., statistical significance of differences between the groups for each variable) as well as individual fold-changes are calculated, and the −log_10_(*p*-value) is then plotted against the log_2_(fold-change) as shown in Figure 9, Figure 10, Figure 11 and Figure 12. The resulting volcano plot can then be used as a tool to visualize the variables with the highest relevance to distinguish between the two sample groups compared [22].

#### 2.5.1. Differences Between the Leaves and Twigs of *Pachysandra terminalis*

A *t*-test was calculated separately for samples from the two different populations to determine significant differences between plant organs, i.e., leaves and twigs. The two volcano plots are shown in Figure 9 and Figure 10.

Buckets representing leaf compounds occur to the far right on the x-axis of the diagram (high values of the fold-change), whereas those occurring mainly in the twigs are on the left side of the x-axis (high negative values of the fold-change). The differences described through the fold-change are significant for buckets located high up on the y-axis (−log_10_(*p*-value ≤ 0.05) ≥ 1.30). Thus, compounds represented by buckets in the upper left and right quadrants of the diagram will be important markers for the differences between the sample groups compared. Epipachysandrine A [18] (12.21 min:*m/z* 467.3666), which was already highlighted in the loadings plot of PC2 vs. PC1, also appeared in the volcano plots as a characteristic compound of the leaves of the *P. terminalis*. The difference is significant between the two groups (leaves and twigs) for both locations (−log_10_(*p*-value) 16.68 for location B and 24.43 for location A). Another bucket showed up as characteristic of the leaves (11.84 min:*m/z* 523.3931), which could not be unambiguously identified. The Reaxys database search in this case yielded two possible hits: Pachysandrine A [2] and (+)-(20*S*)-20-(Dimethylamino)-3α-(methylbenzoylamino)-5*α*-pregn-12*β*-ylacetate [23]. Pachysamine L [24] (12.40 min:*m/z* 485.3764) could be identified as a characteristic compound of the twigs of *P. terminalis*. Another bucket (12.39 min:*m/z* 459.3974) characteristic of the twigs of the plants from location B could not be identified unambiguously. The database search revealed two possible compounds: Pachystermine B [18] or (+)-(20*S*)-20-(Dimethylamino)-3*α*-(methylsenecioylamino)-5*α*-pregn-12*β*-ol [23]. Another compound (12.04 min:*m/z* 979.7712), which could not be identified, is also characteristic of the twigs. It is most likely not an aminosteroid as it showed none of the characteristic core fragments (see Section 2.2.) in the MS/MS spectrum.

#### 2.5.2. Differences Between Plants of Two *Pachysandra terminalis* Populations

To determine differences between plant populations from two different locations, separate *t*-tests were calculated for their respective leaves and twigs. The results were visualized in two volcano plots (see Figure 11 and Figure 12, respectively). Pachystermine C [19] (12.11 min:*m/z* 475.3928), which already showed up in the loadings plot of PC2 vs. PC1, also appeared in the volcano plot of the leaves and the twigs as a characteristic compound of plants from location B. Another aminosteroid (7.00 min:*m/z* 332.2591), which could not be clearly identified, is also characteristic of the leaves and twigs of the plants from location B. Two other aminosteroids were mainly found in the twigs of *P. terminalis* plants from location B. One of these buckets (12.39 min:*m/z* 459.3974) could be identified as a *Pachysandra* alkaloid. The Reaxys database search gave two possible hits: (+)-(20*S*)-20-(dimethylamino)-3*α*-(methylsenecioylamino)-5*α*-pregn-12*β*-ol, which was isolated by Chang et al. from *Pachysandra procumbens* [23], and pachystermine B, which was isolated by Tomita et al. from *P. terminalis* [18]. The bucket (4.95 min:*m/z* 483.3622) was identified as a characteristic compound of the leaves and twigs of plants from location A. The accurate mass, molecular formula, and the characteristic fragments in the MS^2^ spectrum were consistent with those of the previously isolated 2*β*,3*β*,4*β*-Diapachysamine K (**18**) [9], but the retention time was slightly different (*t*_R_(**18**) = 5.37 min). The bucket, therefore, probably represents a constitutional or stereoisomer of 2*β*,3*β*,4*β*-Diapachysamine K, which we tentatively assigned to represent Pachysamin K [24], previously isolated by Sun et al. from *Pachysandra axillaris*. Furthermore, two *Pachysandra* alkaloids previously isolated by us, epipachysamine B (**4**) and pactermine A (**5**), were identified as characteristic compounds of the leaves from location A (from which they were indeed isolated in the previous work) [9]. Another not further identified aminosteroid (6.79 min:*m/z* 463.3913) was also characteristic of the leaves obtained from this location. The twigs of the plants from location A differ in a bucket (4.42 min:*m/z* 432.3587) that was attributed to an (unidentified) aminosteroid due to the characteristic core fragment at *m/z* 285. An especially high log_2_(fold-change) of 4.8 in the leaves and 6.4 in the twigs highlighted the bucket (14.14 min:*m/z* 414.3612). As the corresponding fragment spectrum showed no characteristic core fragment for aminosteroids (see also Figure 3), it could not be further characterized.

## 3. Materials and Methods

### 3.1. Preparation of the Plant Material

Plant material of *Pachysandra terminalis* was collected each month for one year (February 2023–January 2024). To compare the plant material of two different populations, whole aerial parts (shoots with leaves and, if present, flowers) were collected from plants growing in two nearby locations within the city of Münster, Germany (straight-line distance about 700 m): Location A: Botanical Garden, Münster (Germany) (51.96308° N 7.60919° E) and location B: Besides the P + R parking, Coesfelder Kreuz, Münster (Germany) (51.96653° N 7.60115° E) (see Appendix A). The plants were identified by the authors, and voucher specimens are deposited at the Institute of Pharmaceutical Biology and Phytochemistry (IPBP), University of Münster, Germany (voucher No.: IPBP 883 (TS_PT_02), IPBP 884 (TS_PT_03)). Besides the seasonal variations, the variability of alkaloids in the different plant organs, leaves, and twigs should be analyzed as well. Flowers of plants from location B were additionally obtained during the flowering months. The plant material was dried in a drying cabinet (ED 23, Binder GmbH, Tuttlingen, Germany) at 35 °C for 7 days. Afterwards, the leaves, flowers, and twigs were separated. The dried plant material was milled (M20, IKA-Werke GmbH & Co., KG, Staufen, Germany), and 5 g of the powdered material was extracted with 50 mL of ethanol (aqueous, 75%) three times by maceration. The solvent was evaporated under reduced pressure, and the dried extracts were prepared for LC/MS analysis. In total, 50 extracts were obtained (24 leaf extracts, 24 twig extracts, 2 flower extracts).

### 3.2. UHPLC/+ESI-QqTOF-MS/MS-Analysis

All of the obtained extracts were analyzed using an ultra-high-performance liquid chromatograph (Dionex Ultimate RS 3000, Thermo Fisher Scientific, Waltham, MA, USA) coupled with a Bruker micrOTOF-Q II (Bruker Daltonics GmbH, Bremen, Germany) mass spectrometer. The column oven was set to 40 °C, and the flow rate was 0.4 mL/min. As the mobile phase, H_2_O (+0.1% formic acid, A) and Acetonitrile (+0.1% formic acid, B) were used with the following gradient: −0.880 min 15% B, −0.480 min 15% B, 1.000 min 30% B, 7.000 min 33% B, 9.020 min 50% B, 9.050 min 100% B, 15.000 min 100% B, 15.100 min 15% B, 20.000 min 100% B. The injection volume was 2 µL. The dried extracts were dissolved in methanol at a concentration of 10 mg/mL. Papaverin (0.0025 mg/mL) was used as an internal standard. For quality control, 20 µL of each sample was mixed in a LC/MS vial (QC_mix_). As another quality control, the sample PT_GB_B06/23 (*P. terminalis*, Botanical Garden (Münster), leaves, June 2023) was selected (QC). The two quality controls were repeatedly measured after 10 samples to detect any loss of sensitivity or inconsistencies in the bucket alignment. To detect any carry-over of analytes, a blank (methanol) was measured as well after every 20 samples. In the beginning, some of the samples were analyzed to saturate the matrix and thereby obtain stable retention times. To identify interesting compounds, MS^2^ spectra were recorded during this first set of analyses using the auto MS/MS mode (*m/z* 200–1500) with a collision energy of 40 eV. After the matrix saturation, the full set of samples of the seasonal profile of *P. terminalis* was analyzed. All of the following measurements were made without intermediate recording of fragment spectra (i.e., each scan as a full spectrum) so as to obtain the largest possible number of data points for each peak. The samples were first measured consecutively in chronological order. Then, they were analyzed a second time in a random sequence. Thus, each sample is represented twice in the resulting data matrix, further evaluated using multivariate data analysis.

### 3.3. Multivariate Data Analysis

The LC/MS data of the 130 sample measurements were transferred to MetaboScape (V3.0, Bruker Daltonics, Bremen, Germany). In MetaboScape, the molecular features of the chromatograms were extracted, and a bucket table was generated. For further analysis, the following groups were defined:

Organ: Leaves, twigs, flowers

Locations A and B: Botanical Garden, Münster (Germany), P + R Coesfelder Kreuz, Münster (Germany), respectively.

Month: January–December

The filter parameters in MetaboScape were set as follows. Minimum features for extraction: 31/130, Presence of features: 31/130, Filter features by occurrence in groups: Location 15%. The peak detection parameters were set as follows. Intensity threshold: 1000 counts, minimum peak length: 8 spectra, minimum peak length (recursive): 7 spectra, retention time range: 1–15 min, mass range: *m/z* 50–1500. The parameters for ion deconvolution were set as follows. EIC correlation: 0.8, primary ions: [M + H]^+^, seed ions: [M + 2H]^2+^, [M + Na]^+^, [M + K]^+^, common ions: [M + H−-H_2_O]^+^. For the following statistical calculations, the bucket table (1286 bucket variables x 130 analyses) was imported to ProfileAnalysis (V2.1, Bruker Daltonics, Bremen, Germany).

### 3.4. Principal Component Analysis (PCA)

Each bucket of the bucket table was normalized by the internal standard papaverine (4.46 min:*m/z* 340.1553). Principal component analyses were calculated with the unscaled data as well as with different scaling methods: unit variance, Pareto, and level scaling. The results were relatively similar with all methods (see Appendix A), yielding a good separation of the samples and thus reflecting the major differences between their variables/compounds. The Pareto-scaled model was chosen for further analysis for the reasons pointed out in 2–4.

The calculated PCA models were all validated by the leave-one-out cross-validation scheme.

### 3.5. Volcano Plot

To compare two of the previously defined groups based on the bucket table, a *t*-test was calculated in ProfileAnalysis and afterwards visualized as a volcano plot. Buckets with a high fold-change (log_2_(∣fold-change∣ ≥ 2.5) ≥ 1.3) and a low *p*-value (−log_10_(*p*-value ≤ 0.05) ≥ 1.30) were considered for evaluation.

## 4. Conclusions

With multivariate data analysis (PCA and volcano plots), changes in the alkaloid profile of *Pachysandra terminalis* throughout the vegetation period, as well as in the different plant organs and in plants from two different locations in Münster (Germany), could be characterized. PCA revealed that the alkaloid profile differed mainly between plant organs (PC1, 45% variance explained). The differences between the plants from the two locations could be described with the second principal component (PC2, 16%), and the seasonal variations accounted for only 7% (PC3) of the total variance of the data set.

The large difference between plant organs (PC1) could be explained by the fact that plants use alkaloids mainly for defense against herbivores and microbes, so it appears straightforward that alkaloids of different structures and bioactivity will accumulate in different plant organs, which are likely to be attacked by different pests.

We chose to investigate two different populations from distinct but geographically close locations in order to elucidate whether factors independent of the weather and climatic change over the vegetation period also cause relevant differences in chemical profiles. These could be external differences between the two locations, such as sunlight/shadow, soil composition, as well as internal (i.e., genetic) differences between the two populations. Detailed elucidation of the exact nature of such influences being beyond the scope of this present work, we actually aimed at finding out how strong they can be in comparison with the seasonal changes. Apparently, the influence of such factors is quite significant since they define the second principal component, PC2, i.e., the second largest fraction of variance in the data, and are thus more influential than the changes over time defining PC3, which are determined by climate and weather.

The evaluation of the scores and loadings plots, as well as the volcano plots, revealed 25 compounds (see Appendix A, summarizing all these compounds’ features) that were characteristic of the plant organs, plants from one of the locations, or the season. To obtain detailed explanations for the observed differences, many other factors need to be analyzed and monitored (e.g., differences in the nutrient or water content of the soil). It should also be acknowledged that plants from only two very close locations were selected for this first study, for the reason mentioned above. To obtain a further understanding of the variability of the alkaloid profile of *P. terminalis* due to ecological factors, it would be necessary to investigate the differences between plants from distant populations and certainly a wider variety of populations. Nevertheless, the current study showed that the alkaloid profile in different parts of *P. terminalis*, from different locations and collected at different times, may vary significantly. Careful selection of plant material is therefore essential for the isolation of specific compounds, e.g., with a particular desired pharmacological effect, from this plant. The present work will be a useful guide for such future investigations, which may address the question of which mechanisms and internal as well as external factors in detail influence the enzyme expression and activity along the biosynthetic pathways leading to the great diversity of alkaloids in *P. terminalis*.

## Figures and Tables

**Figure 1 plants-14-03060-f001:**
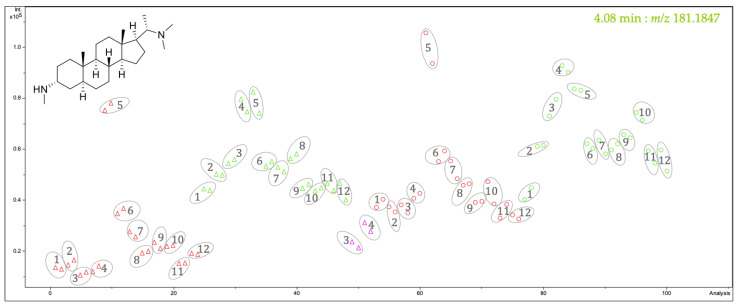
Bucket statistic view of the bucket identified as pachysamine A (**8**) (4.08 min:*m/z* 181.1847). The samples from location A are displayed as dots, and the samples from location B are displayed as triangles. Samples of the twigs are green, and those of the leaves are red. The pink triangles belong to samples of the flowers from *P. terminalis* of location B. The numbers indicate the month from January (1) to December (12). Each sample was analyzed twice, and the results were plotted individually but circled together.

**Figure 2 plants-14-03060-f002:**
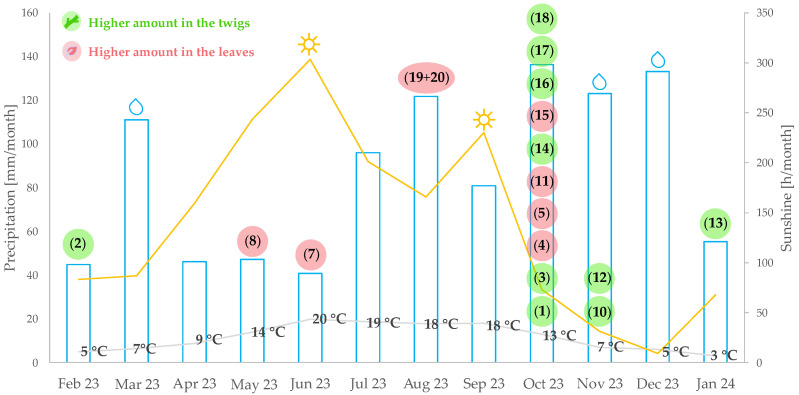
Optimum harvesting periods for the previously isolated *Pachysandra* alkaloids for the plants from the Botanical Garden (Münster) throughout the year (February 2023 to January 2024), as well as the weather data for each month provided by the German Weather Service “Deutscher Wetterdienst”, station Münster/Osnabrück (ID1766) [14]. Numerals in parentheses are compound numbers as used in the text and in Table 1.

**Figure 3 plants-14-03060-f003:**
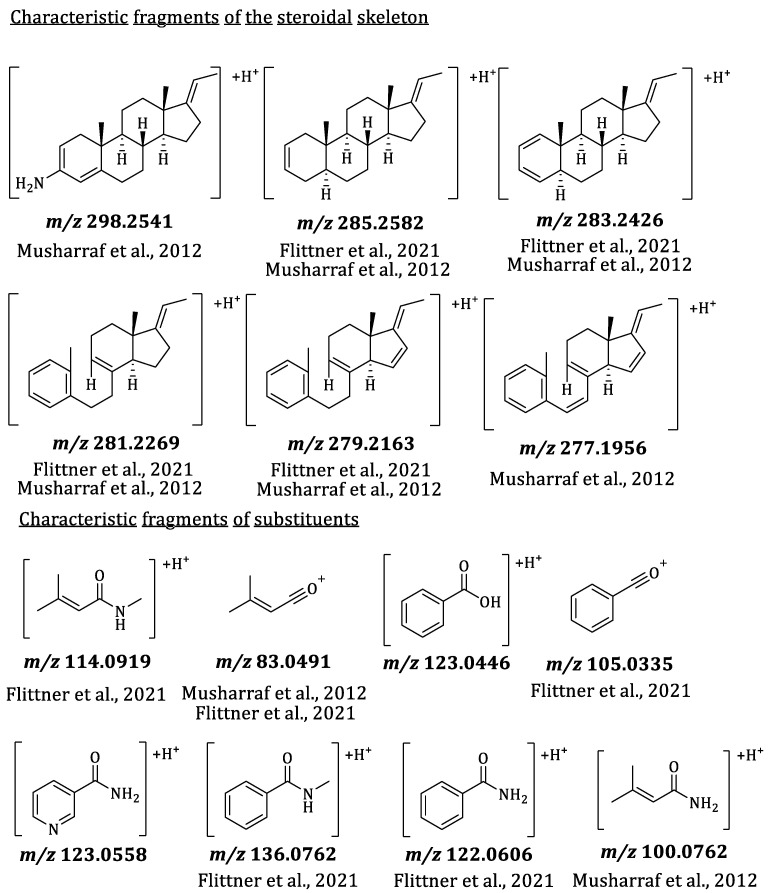
Characteristic MS/MS fragments of the steroidal skeleton and from substituents of aminosteroids from *Sarcococca* and *Pachysandra*. Most fragments were already described by Musharraf et al., 2012 [15] and Flittner et al., 2021 [8] before.

**Figure 4 plants-14-03060-f004:**
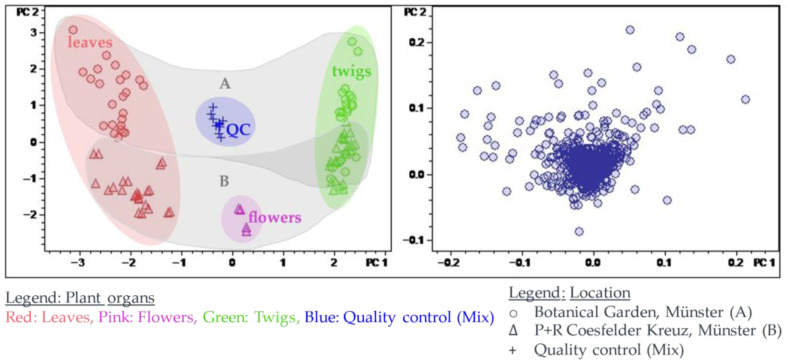
Principal component analysis of the LC/MS data of the seasonal profile of *Pachysandra terminalis* with Pareto scaling. Scores plot left, loadings plot right. Plots show PC2 (16% variance explained) plotted vs. PC1 (45% variance explained). The relevant buckets for the clustering in the scores plot are identified from the loadings plot shown in Figure 5.

**Figure 5 plants-14-03060-f005:**
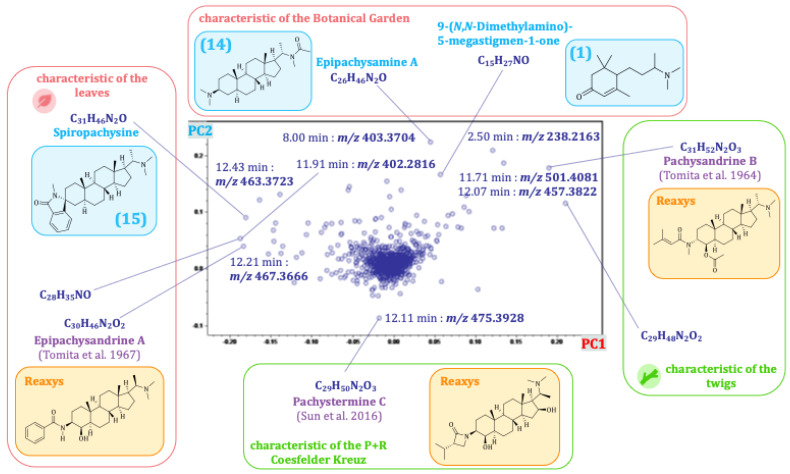
Loadings plot (PC2 vs. PC1) of the principal component analysis of the seasonal profile of *Pachysandra terminalis*. The compounds highlighted in orange were identified by a search of the Reaxys database on the basis of their molecular mass, those in blue were previously isolated by us [9] and could be identified by direct comparison. The references cited in the figure are Tomita et al., 1964 [2], Tomita et al., 1967 [18] and Sun et al., 2016 [19].

**Figure 6 plants-14-03060-f006:**
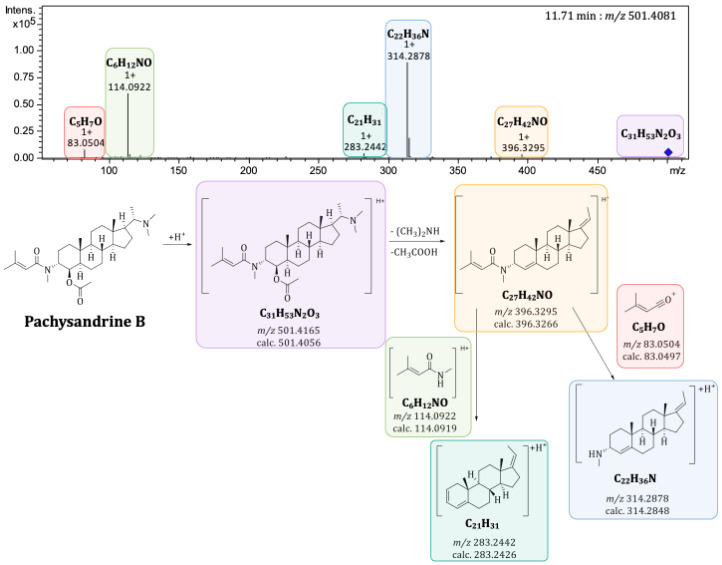
+ESI-QqTOF-MS/MS spectrum of the bucket (11.71 min:*m/z* 501.4081) identified as pachysandrine B and the postulated fragment pattern.

**Figure 7 plants-14-03060-f007:**
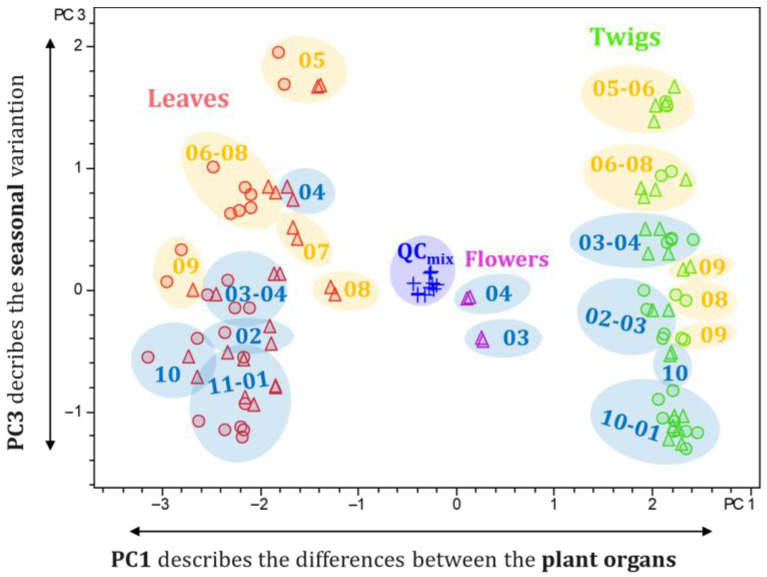
Scores plot (PC3 vs. PC1) of the principal component analysis of the seasonal profile of *Pachysandra terminalis*. Cold and warm months (definition see text) are highlighted in blue and yellow. The numbers from 01 to 12 correspond to the months from January to December.

**Figure 8 plants-14-03060-f008:**
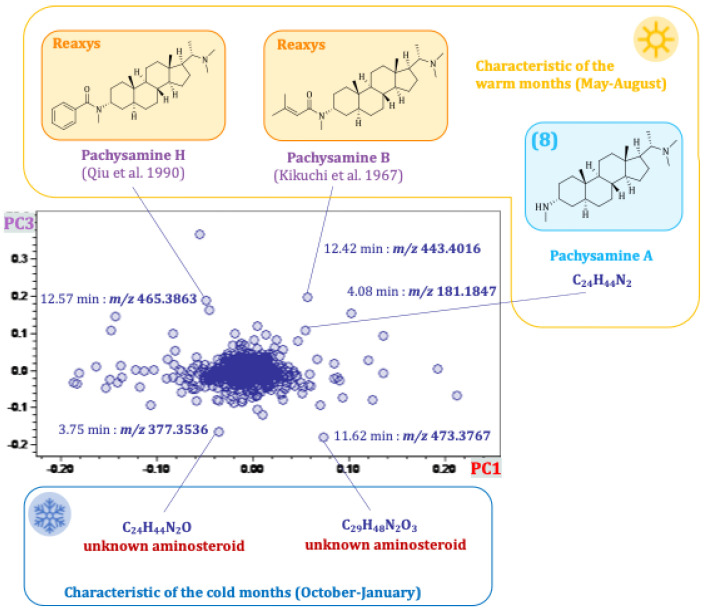
Loadings plot (PC3 vs. PC1) of the principal component analysis of the seasonal profile of *Pachysandra terminalis*. The compounds highlighted in orange were identified based on their molecular mass through a Reaxys database search, those in blue were previously isolated by us [9] and could be identified by direct comparison. The references cited in the figure are Qiu et al., 1990 [20] and Kikuchi et al., 1967 [21].

**Figure 9 plants-14-03060-f009:**
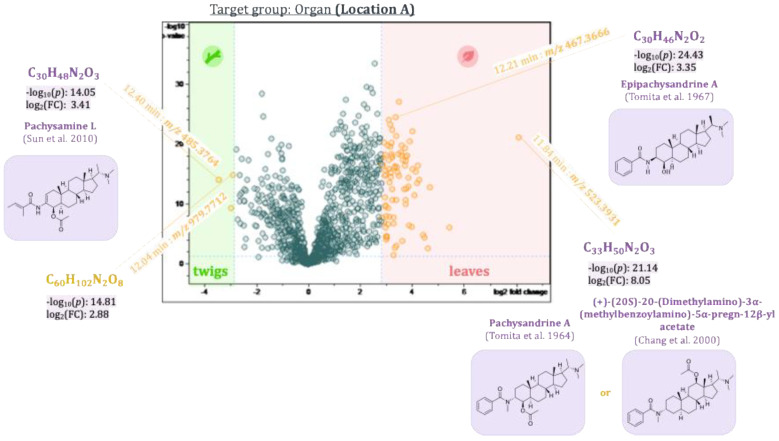
Volcano plot of the samples from the leaves and twigs from *Pachysandra terminalis* from location A. Buckets highlighted in yellow differ between the two groups with a great fold-change (log2(∣fold-change∣ ≥ 7.0) ≥ 2.8), and that difference is significant (−log10 (*p*-value ≤ 0.05) ≥ 1.30). Some buckets could be identified as *Pachysandra* alkaloids based on their accurate mass, molecular formula, and fragment spectra. The references cited in the figure are Tomita et al., 1964 [2], Tomita et al., 1967 [18], Chang et al., 2000 [23] and Sun et al., 2010 [24].

**Figure 10 plants-14-03060-f010:**
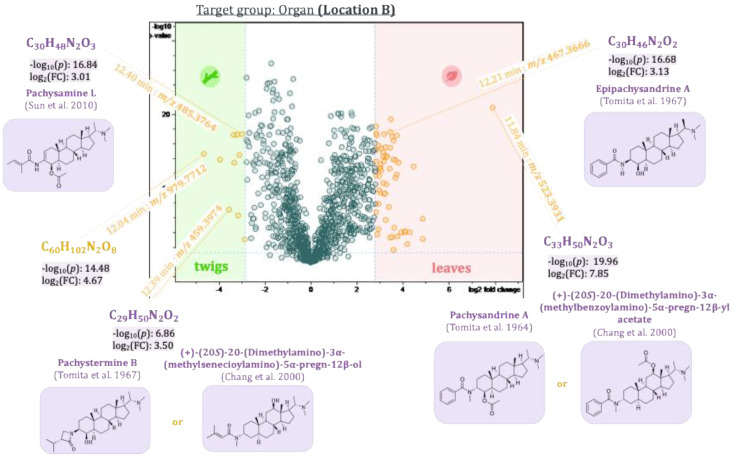
Volcano plot of the samples from the leaves and twigs from *Pachysandra terminalis* from location B. Buckets highlighted in yellow differ between the two groups with a great fold change (log2(∣fold change∣ ≥ 7.0) ≥ 2.8), and that difference is significant (−log10 (*p*-value ≤ 0.05) ≥ 1.30). Some buckets could be identified as *Pachysandra* alkaloids based on their accurate mass, molecular formula, and fragment spectra. The references cited in the figure are Tomita et al., 1964 [2], Tomita et al., 1967 [18], Chang et al., 2000 [23] and Sun et al., 2010 [24].

**Figure 11 plants-14-03060-f011:**
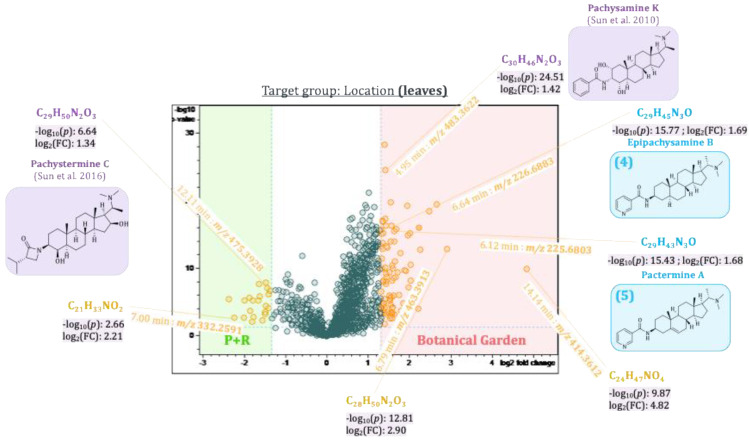
Volcano plot of the samples from the two different locations from leaves of *Pachysandra terminalis*. Buckets highlighted in yellow differ between the two groups with a great fold-change (log2(∣fold-change∣ ≥ 2.5) ≥ 1.3), and that difference is significant (−log10 (*p*-value ≤ 0.05) ≥ 1.30). Some buckets could be identified as *Pachysandra* alkaloids based on their accurate mass, molecular formula, and fragment spectra. The references cited in the figure are Sun et al., 2016, 2010 [19, 24].

**Figure 12 plants-14-03060-f012:**
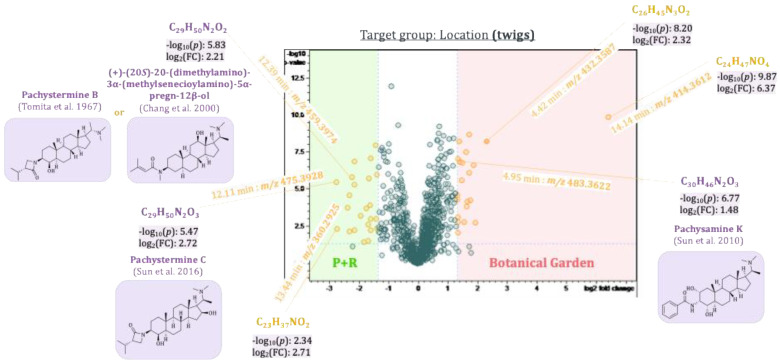
Volcano plot of the samples from the two different locations from twigs of *Pachysandra terminalis*. Buckets highlighted in yellow differ between the two groups with a great fold-change (log2(∣fold-change∣ ≥ 2.5) ≥ 1.3), and that difference is significant (−log10 (*p*-value ≤ 0.05) ≥ 1.30). Some buckets could be identified as *Pachysandra* alkaloids based on their accurate mass, molecular formula, and fragment spectra. The references cited in the figure are Tomita et al., 1967 [18], Sun et al., 2016 [19], Chang et al., 2000 [23] and Sun et al., 2010 [24].

**Table 1 plants-14-03060-t001:** Optimum harvesting periods for the *Pachysandra* alkaloids previously isolated by our group [9].

Alkaloid	Bucket <*t*_R_:*m/z*>	Type of Ion	Optimum Harvesting Period	Location	Organ
9-(*N*,*N*-Dimethylamino)-5-megastigmen-1-one (**1**)	2.50 min:*m/z* 238.2163	[M + H]^+^	October	A > B	twigs > leaves
5,6-Dehydro-desacyl-epipachysamine A (**2**)	2.78 min:*m/z* 180.1789	[M + 2H]^2+^	January, February, May	A > B	twigs ≈ leaves
Desacyl-epipachysamine A (**3**)	3.28 min:*m/z* 181.1848	[M + 2H]^2+^	October, November	A > B	twigs > leaves
Epipachysamine B (**4**)	6.64 min:*m/z* 226.6883	[M + 2H]^2+^	October	A > B	A: leaves > twigs B: twigs > leaves
Pactermine A (**5**)	6.12 min:*m/z* 225.6803	[M + 2H]^2+^	October	A > B	A: leaves >> twigs B: twigs ≈ leaves
3α,4α-Diapachsanaximine A (**7**)	5.24 min:*m/z* 481.3444	[M + H]^+^	June	A > B	leaves > twigs
Pachysamine A (**8**)	4.08 min:*m/z* 181.1847	[M + 2H]^2+^	April, May	A > B	twigs > leaves
Pachysandrine D (**10**)	5.74 min:*m/z* 459.3918	[M + H]^+^	October, November	A > B	twigs > leaves
Terminaline (**11**)	4.32 min:*m/z* 364.3219	[M + H]^+^	October	A > B	twigs ≈ leaves
*N*-methyl-desacyl-epipachysamine A (**12**)	3.21 min:*m/z* 188.1925	[M + 2H]^2+^	October–December	A > B	twigs >> leaves
Sarcodinine (**13**)	2.69 min:*m/z* 187.1832	[M + 2H]^2+^	January–March	A ≈ B	twigs > leaves
Epipachysamine A (**14**)	8.00 min:*m/z* 403.3704	[M + H]^+^	October	A > B	twigs > leaves
Spiropachysine (**15**)	12.43 min:*m/z* 463.3723	[M + H]^+^	October	A ≈ B	twigs < leaves
4β-Hydroxy-hookerianamide N (**16**)	5.67 min:*m/z* 461.3757	[M + H]^+^	October	A > B	twigs > leaves
5α-Hydroxy-3α,4α-diapachysanaximine A (**17**)	5.49 min:*m/z* 249.1928	[M + 2H]^2+^	October	A > B	twigs > leaves
2β,3β,4β-Diapachysamine K (**18**)	6.54 min:*m/z* 483.3607	[M + H]^+^	October	A > B	twigs > leaves
3β-Dimethylamino-pregnane-20-oxime (**19**)	8.98 min:*m/z* 361.3224	[M + H]^+^	June–August	A > B	twigs ≈ leaves
3β-Dimethylamino-pregn-5,6-ene-20-oxime (**20**)	8.05 min:*m/z* 359.3070	[M + H]^+^	June–August	A > B	twigs ≈ leaves

## Data Availability

The original contributions presented in this study are included in the article/Appendix A. The raw data supporting the conclusions of this article will be made available by the authors on request.

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
