# Peer review of "Seasonal, Organ-, and Location-Dependent Variations in the Alkaloid Content of Pachysandra terminalis Investigated by Multivariate Data Analysis of LC-MS Profiles"

_plants, 2025, doi:10.3390/plants14193060_

Round 1
Reviewer 1 Report
Comments and Suggestions for Authors
In this manuscript, Lizanne Schäfer and colleagues investigated variations in the alkaloid profile of P. terminalis related to seasonal changes as well as differences between plant organs (leaves and twigs) and between plant populations. I have following comments:
1, For the Title, it is better to replace the word “location” with “ecotype”.
2, For the Abstract section, main scientific questions answered by this study should be described, and practical interests of this study should be stated.
3, For the keywords, authors should minimize the keywords
4, For the introduction section, full name of the abbreviation “UHPLC/+ESI-QqTOF-MS/MS” should be spelt out.
5, For the results, pictures showing P. terminalis organs examined in this study should be exhibited, and error bars should be presented in the Figure 2.
6, For the Materials and methods, experimentation site should be presented on a map, and genotype of plant samples should be clarified. Method for the statistical analysis should be described.
7, An independent discussion section should be presented.
Author Response
Reviewer 1
In this manuscript, Lizanne Schäfer and colleagues investigated variations in the alkaloid profile of P. terminalis related to seasonal changes as well as differences between plant organs (leaves and twigs) and between plant populations. I have following comments:
1, For the Title, it is better to replace the word “location” with “ecotype”.
We are sorry but we do not agree: It is not clear whether these two populations represent different ecotypes, but it is a fact that they grow at different locations.
2, For the Abstract section, main scientific questions answered by this study should be described, and practical interests of this study should be stated.
In our opinion, the abstract contains the main scientific questions and answers. In other words: “Which compounds are the main determinants of differences in the chemical profiles of the investigated plant samples over the vegetation time?” However, we have more clearly specified the main outcome of the study by modifying the last part of the absctact:
“The results of these analyses, for the first time, allowed a holistic analysis of variation in the alkaloid profiles in P. terminalis organs over the vegetation period. The evaluation of the PCA scores and loadings plots of principal components 1 through 3 as well as of volcano plots highlighted 25 different compounds, mostly identified as aminosteroid alkaloids, that were most relevant for the differences between leaves and twigs and between the two populations and mainly determined the changes in their chemical profiles over the vegetation period.”
3, For the keywords, authors should minimize the keywords
We find the keywords quite appropriate reflecting the manuscript’s contents. Maybe the reviewer could be more specific advising us which keywords are unnecessary?
4, For the introduction section, full name of the abbreviation “UHPLC/+ESI-QqTOF-MS/MS” should be spelt out.
We agree with the reviewer and this was done.
5, For the results, pictures showing P. terminalis organs examined in this study should be exhibited, ..
We agree with the reviewer and have added pictures showing the plants of the two locations more closely, as well as photos of the voucher specimens, showing the organs more clearly.
.. and error bars should be presented in the Figure 2.
Error bars cannot be added to Fig. 2, because there are none. In fact, two analyses of each sample are shown separately in each case, and they are encircled showing the closeness in all cases. This has been clarified in the figure caption (“Each sample was analyzed twice and the results plotted individually but circled together.”)
6, For the Materials and methods, experimentation site should be presented on a map, and genotype of plant samples should be clarified.
A map showing both locations has been added to Figure S1.
We are sorry but the genotype was not determined since we have no possibility to do this at our laboratory. It is also not a mandatory requirement (yet?) in phytochemical research to do this, as long as samples or vouchers of the investigated organisms are kept. We can provide samples of the plant material at any time.
Method for the statistical analysis should be described.
In our opinion, all statistical methods are described in full detail. Maybe the reviewer would like to advise us specifically which aspects are missing?
7, An independent discussion section should be presented.
Although the terms “Results” and “Discussion” are mentioned separately in the authors’ instructions, we are not aware that it is a mandatory prescription by the journal to have separate sections. In fact, we find it appropriate to discuss the results directly along with their description. Manuscript length is frequently unnecessarily expanded and the information content rendered partly redundant by separate discussion sections since the results are often repeated. We hope the editor will allow us to keep the current manuscript structure. Otherwise, we would have to re-write the whole paper.
We thank the reviewer for the very constructive criticism and the time and effort spent to help us improve our manuscript!

Reviewer 2 Report
Comments and Suggestions for Authors
The manuscript presents a study on the seasonal variation of alkaloid content in two selected populations of Pachysandra terminalis. The topic is relevant, and the data are of potential interest; however, several issues should be addressed before the manuscript can be considered for publication.
1. Figure S1 – Image Quality and Clarity:
Figure S1 appears to be too small to allow a clear visualization of the plant. It is recommended that the authors provide a high-resolution, enlarged image of the entire P. terminalis plant. The authors should also provide an image that clearly indicates the different plant parts (e.g., leaves, twigs, flowers, etc.) that were used for alkaloid analysis. Providing this context is essential to better understand the anatomical source of the alkaloids and to aid in future replication of the study.
2. Selection of Sampling Sites:
While the LC-MS analysis is robust and well-conducted, the authors limited their sampling to only two populations of P. terminalis, both located in close geographic proximity. It is unclear why these particular locations were chosen for comparison. The authors should clarify their rationale for selecting these specific populations. Were they chosen due to accessibility, observed morphological differences, or known phytochemical variability?
Moreover, sampling from populations that are geographically distant or growing under different environmental conditions (e.g., altitude, soil type, light exposure) would have provided a more comprehensive understanding of how alkaloid profiles vary in response to ecological factors. The limited geographic scope should be acknowledged in the discussion section as a limitation of the study, and the authors are encouraged to suggest future investigations that include more diverse and distant populations for broader chemogeographic comparisons.
3. Comparative Alkaloid Profiling and Biosynthesis Consideration:
The manuscript would benefit from a conclusive summary of the characteristic alkaloids identified in the different plant organs across the two populations. Specifically, the authors should highlight the major alkaloids detected in the leaves, twigs, and other organs, and comment on any population-specific differences in their distribution.
Additionally, the discussion should explore the potential correlation between the observed alkaloid profiles and the biosynthetic pathway of aminosteroid alkaloids known to be present in P. terminalis. For instance, differences in the expression of biosynthetic enzymes or precursor availability in different tissues or seasons could explain the variations in alkaloid levels. Such an analysis would strengthen the biochemical relevance of the findings and offer deeper insights into the metabolic regulation in this species.
Author Response
Reviewer 2
The manuscript presents a study on the seasonal variation of alkaloid content in two selected populations of Pachysandra terminalis. The topic is relevant, and the data are of potential interest; however, several issues should be addressed before the manuscript can be considered for publication.
- Figure S1 – Image Quality and Clarity:
Figure S1 appears to be too small to allow a clear visualization of the plant. It is recommended that the authors provide a high-resolution, enlarged image of the entire P. terminalisplant. The authors should also provide an image that clearly indicates the different plant parts (e.g., leaves, twigs, flowers, etc.) that were used for alkaloid analysis. Providing this context is essential to better understand the anatomical source of the alkaloids and to aid in future replication of the study.
The figure was mainly meant to show the different locations of the two population rather than the plant itself. However, we agree with the reviewer that it is useful to show the plant as such. We have therefore added two additional photograph panels for each accession showing mor closely the two plant populations as well as the dried herbarium specimens of single shoots, in which also the twigs and leaves are more easy to see.
- Selection of Sampling Sites:
While the LC-MS analysis is robust and well-conducted, the authors limited their sampling to only two populations of P. terminalis, both located in close geographic proximity. It is unclear why these particular locations were chosen for comparison. The authors should clarify their rationale for selecting these specific populations. Were they chosen due to accessibility, observed morphological differences, or known phytochemical variability?
The reviewer is right, this should be explained more clearly. Indeed, beside equally good accessibility and sufficient abundance (sample collection every month over a whole year), we wanted to investigate whether chemical differences occur between two populations of the same species related to factors independent of climate/weather, which vary over the vegetation period. Climate/weather are identical in two spots so close together, but soil, shadow/irradiation and other external factors may vary. What may also vary, of course, is the genotype, which we could not distinguish. Without being able to study such influences in detail, we actually aimed at finding out how strong they are on an overall scale. Apparently, they are quite significant since they define the second principal component of the PCA, i.e. the second largest fraction of variance in the data, and are hence more influential than the changes over time (determined by climate/weather), which only make up for the third component. We tried to clarify this and mention that further, more detailed studies would have to be conducted to find out which are the exact determinants for this part of the variation which we currently can only attribute to the fact that two populations from distinct locations were studied. The statement explaining this has been added to the conclusions section (3rd paragraph).
Moreover, sampling from populations that are geographically distant or growing under different environmental conditions (e.g., altitude, soil type, light exposure) would have provided a more comprehensive understanding of how alkaloid profiles vary in response to ecological factors. The limited geographic scope should be acknowledged in the discussion section as a limitation of the study, and the authors are encouraged to suggest future investigations that include more diverse and distant populations for broader chemogeographic comparisons.
The reviewer is absolutely right. A statement acknowledging this and stressing the necessity for further studies with a higher number of geographically more remote populations was also added to the conclusion.
- Comparative Alkaloid Profiling and Biosynthesis Consideration:
The manuscript would benefit from a conclusive summary of the characteristic alkaloids identified in the different plant organs across the two populations. Specifically, the authors should highlight the major alkaloids detected in the leaves, twigs, and other organs, and comment on any population-specific differences in their distribution.
We hope we may draw the reviewer’s attention to table S1 in the supplementary materials, where exactly this requested information is comprehensively summarized. Since this information is very complex, we believe it is better summarized in tabular form than in a descriptive text, which would extend the length without being more informative. A somewhat more conspicuous hint to this important table was added in the text (Conclusions, first sentence in last paragraph).
Additionally, the discussion should explore the potential correlation between the observed alkaloid profiles and the biosynthetic pathway of aminosteroid alkaloids known to be present in P. terminalis. For instance, differences in the expression of biosynthetic enzymes or precursor availability in different tissues or seasons could explain the variations in alkaloid levels. Such an analysis would strengthen the biochemical relevance of the findings and offer deeper insights into the metabolic regulation in this species.
The reviewer is certainly right in anticipating that this would be very interesting information. However, since there is no experimental evidence on specific biosynthetic sequence, the necessary enzymes and their expression levels, construction of such biosynthetic explanations would inevitably be (a) speculative and (b) very complex to describe. We would rather avoid such speculation and leave this as an interesting topic for future studies. Instead, a short statement indicating this was added (last sentence of conclusions), which we hope will satisfy the reviewer.
We thank the reviewer for the very constructive criticism and the time and effort spent to help us improve our manuscript!

Reviewer 3 Report
Comments and Suggestions for Authors
This study is a continuation of other studies conducted by the same authors on P. terminalis. This study is, therefore, a step forward in the search for compounds with antiprotozoal activity. So, it's a job well done that doesn't require any further comments.
Author Response
Reviewer 3
This study is a continuation of other studies conducted by the same authors on P. terminalis. This study is, therefore, a step forward in the search for compounds with antiprotozoal activity. So, it's a job well done that doesn't require any further comments.
We thank the reviewer for the positive assessment.

Reviewer 4 Report
Comments and Suggestions for Authors
There is a major problem in terms of the consistency of the presented mass spectral data as detailed on the attached marked manuscript. The defined discriminating ion peaks do not correspond correctly to the MWS of the targeted metabolites particularly those presented in table 1. For example, I do not understand the use of the ion peak at m/z 181 for pachysamine A and I cannot comprehend whether this is a base peak occurring during fragmentation, which was not as well indicated in Figure 3, which was suppose to present expected fragment ions typical for steroidal alkaloids in this study. However, from section 2.4.1 the representation of these discriminating metabolites were correct. Was the manuscript written by two people and were unable to discuss their results together? The wrong results presented in Sections 2.2 and 2.3 were not coherent with that of Section 2.4, which presented the results correctly. The authors need to look closely on these. Similarly, Table 1 is not coherent with the data presented on the supplementary information.
The PCA results simply indicated a natural separation and correlation of the metabolites according to their plant parts as well as other parameters as generated by the model. The separation of the groups were very clear that it did not require an Orthogonal partial least squares discriminant analysis to determine the discriminating metabolites for each respective groups. However, on the discussion part, the ecological significance between the collection locations were not properly explored.
Further details are needed to be added in the Methodology section such as UPLC gradient used, flow rates, injection volume and fragmentation parameters applied. Please check the concentration used as well. It seems that the concentration of 10 mg/mL is quite high for a UPLC-HRMS method.

The quality of the English Language needs to be improved for better readability. Avoid using parenthesis when the information can be coherently included on the main sentence construction itself.
Author Response
Reviewer 4
There is a major problem in terms of the consistency of the presented mass spectral data as detailed on the attached marked manuscript. The defined discriminating ion peaks do not correspond correctly to the MWS of the targeted metabolites particularly those presented in table 1. For example, I do not understand the use of the ion peak at m/z 181 for pachysamine A and I cannot comprehend whether this is a base peak occurring during fragmentation, which was not as well indicated in Figure 3, which was suppose to present expected fragment ions typical for steroidal alkaloids in this study. However, from section 2.4.1 the representation of these discriminating metabolites were correct. Was the manuscript written by two people and were unable to discuss their results together? The wrong results presented in Sections 2.2 and 2.3 were not coherent with that of Section 2.4, which presented the results correctly. The authors need to look closely on these. Similarly, Table 1 is not coherent with the data presented on the supplementary information.
The mass spectral information used and presented in our work is all correct: The ions used for bucketing are usually the more intense ions which in case of our alkaloids may be either singly protonated and charged [M+H]+ , or, in many cases with two basic amino groups, the [M+2H]2+ species. Thus, in case of pachysamine a, this accounts for (360+2)/2=181. The reviewer is asked to compare the info in Table 1 with the MS data reported for each of the compounds in our previous paper (Reference [9] https://doi.org/10.3390/molecules30051093). In order to avoid that other readers are subject to the same misunderstanding, we have added a column “Type of ion” to the table so that this becomes clear.
Furthermore, we noticed some inconsistencies in the elemental formulas reported in the Figures showing structure assignments along with the compound buckets of PCA and volcano plots. In some instances, formulas of charged ions were reported while in others it was the compounds’ neutral elemental formula. Since the structures are depicted in the neutral form, we have modified the elemental formulas so that all represent uncharged, neutral molecules. We hope that this will resolve all questions regarding compound assignment and representation.
The PCA results simply indicated a natural separation and correlation of the metabolites according to their plant parts as well as other parameters as generated by the model. The separation of the groups were very clear that it did not require an Orthogonal partial least squares discriminant analysis to determine the discriminating metabolites for each respective groups. However, on the discussion part, the ecological significance between the collection locations were not properly explored.
In our paper we used PCA to investigate the differences between the various sample groups. It is a misunderstanding that such methods are only used to discriminate between groups (as the reviewer appears to think), but in fact they are useful tools to study the reasons/determinants for such differences (i.e. answer the question, e.g., “which compounds mainly determine the differences between leaves and twigs?”). Since the solution is often (as in the present case) multifactorial, it is absolutely purposeful to use PCA. In our case, for instance, it could be expected that there would be certain differences occurring over time (third PC), but PCA helped us to understand that the differences between the two locations are even more influential (second PC). It would be absolutely impossible to separate between these two factors (changes over time vs changes between two locations) without using a multivariate model (here PCA was sufficient and a supervised distinguishing method such as OPLS-DA was not necessary, as the reviewer noticed).
Further details are needed to be added in the Methodology section such as UPLC gradient used, flow rates, injection volume and fragmentation parameters applied. Please check the concentration used as well.
The reviewer is right, this was forgotten. We have added the missing information.
It seems that the concentration of 10 mg/mL is quite high for a UPLC-HRMS method.
The concentration is our standard concentration for the analysis of unknown plant extract samples. It ensures that also many minor constituents can be properly detected and identified/derelplicated with meaningful mass spectral data.
[The English could be improved to more clearly express the research.]
We have checked the language once more and made some improvements according to the reviewer’s annotated pdf. We are particularly grateful for the lot of work the reviewer spent on this.
We thank the reviewer for the very constructive criticism and the time and effort spent to help us improve our manuscript!

Round 2
Reviewer 1 Report
Comments and Suggestions for Authors
Authors have answered my questions in the revision.
Author Response
No further revision is requested by this reviewer. Once more, we doe appreciate the time and effort to assess our work!
Reviewer 2 Report
Comments and Suggestions for Authors
The authors have improved the manuscript significantly.
Author Response
No further revision is requested by this reviewer. Once more, we do appreciate the time and effort to assess our work!
Reviewer 4 Report
Comments and Suggestions for Authors
The manuscript was improved as requested. However, please use the terminology doubly charged ions, which they are actually observing other than expected molecular ion peaks.
Otherwise, the manuscript can proceed for publication.
Author Response
The manuscript was improved as requested. However, please use the terminology doubly charged ions, which they are actually observing other than expected molecular ion peaks.
Reply: We are not sure what the reviewer means. But to improve the clarity, we have added a short explanatory statement in the sentence in line 219-221:
"For a closer characterization of the buckets, the molecular formula was obtained from the accurate mass (usually representing the [M+H]+ ion, in case of compounds with two basic amino groups also the doubly charged [M+2H]2+ ion) and compared with the previously isolated Pachysandra alkaloids."
We hope that everything is now explained clearly enough and thank the reviewer, once more, for the time and effort spent to help us improve our work.